# Vacuum-Assisted Block Freeze Concentration Studies in Cheese Whey and Its Potential in Lactose Recovery

**DOI:** 10.3390/foods12040836

**Published:** 2023-02-15

**Authors:** Noelia Gil, Gisela Quinteros, Monica Blanco, Shafirah Samsuri, Nurul Aini Amran, Patrico Orellana-Palma, Elane Schwinden, Eduardo Hernández

**Affiliations:** 1Agri-Food Engineering and Biotechnology Department, Universitat Politècnica de Catalunya BarcelonaTech, Campus del Baix Llobregat, Edifici D-4 C/Esteve Terradas, 8, Castelldefels, 08860 Barcelona, Spain; 2Department of Mathematics, Universitat Politécnica de Catalunya BarcelonaTech, Parc Mediterrani de la Tecnologia Campus del Baix Llobregat, Edifici D-4 C/Esteve Terradas, 8, Castelldefels, 08860 Barcelona, Spain; 3Chemical Engineering Department, Universiti Teknologi PETRONAS, 32610 Seri Iskandar, Perak, Malaysia; 4Departamento de Ingeniería en Alimentos, Facultad de Ingeniería, Campus Andrés Bello, Universidad de La Serena, Avda. Raúl Bitrán 1305, La Serena 1720010, Chile; 5Departamento de Ciência e Tecnologia de Alimentos, Universidade Federal de Santa Catarina, Florianópolis 88036, Santa Catarina, Brazil

**Keywords:** block freeze concentration, vacuum, whey, lactose

## Abstract

Block freeze concentration (BFC) is considered an emerging technology which allows the acquiring of high quality organoleptic products, due to the low temperatures employed. In this study we have outlined how the vacuum-assisted BFC of whey was investigated. The effects of vacuum time, vacuum pressure, and the initial solids concentration in whey were studied. The results obtained show that the three variables significantly affect each of the following parameters analysed: solute yield (Y) and concentration index (CI). The best Y results were obtained at a pressure of 10 kPa, 7.5 °Bx, and 60 min. For CI parameter, the highest values were given at 10 kPa, 7.5 °Bx, and 20 min, respectively. In a second phase, by applying the conditions that provide higher solute yield to three different types of dairy whey, Y values of 70% or higher are reached in a single step, while that the CI of lactose are higher than those of soluble solids. Therefore, it is possible to recover, in a single step, at least 70% of the lactose contained in the initial whey samples. This suggests that vacuum-assisted BFC technology may be an interesting alternative for the recovery of lactose contained in whey.

## 1. Introduction

In Mediterranean countries, where the dairy sector (cheese making-oriented) has a marked traditional character, constraints in the utilization of fresh liquid whey are linked to high transportation costs of the bulky liquid and the low productivity of drying facilities [1]. Thereby, an excellent alternative to reduce operating time, maintenance costs, and storage spaces, among others, is the reduction of the volume (and the weight) of fresh liquid whey. For this purpose, it is important to concentrate these solutions, and thus, this process allows for the decreasing of the water activity, and in turn, to avoid any unwanted microbial growth [2]. Precisely, there are three main methods available to eliminate the part of water contained in objective solutions, allowing concentration in the liquid food products: membrane processes, vacuum evaporation, and freeze concentration. Although these technologies are effective, they also present certain disadvantages. For example, vacuum evaporation technologies are extremely expensive, and it has been found that the energy use is very high. In the same way, membrane technologies needs frequent maintenance because of membrane fouling [3]. Hence, when comparing the heat of evaporation (about 2260 kJ/kg under pressure of 0.1 MPa) with enthalpy of freezing (335 kJ/kg), the freeze concentration process seems to be cheaper than evaporation from the energy point of view.

Specifically, freeze concentration (FC) is considered an emerging technology in which liquid food is concentrated via partial or total water freezing, where the procedure involves a controlled decrease in the temperature of the liquid food. As a result of this process it is pushed below the freezing point, avoiding the eutectic temperature, where all the components of the product are frozen. Hence, FC process separates the ice fraction from the residual unfrozen solution [4]. Additionally, there are three techniques viable to applied freeze concentrate in liquid foods: suspension freeze concentration (SFC), progressive freeze concentration (PFC), and block freeze concentration (BFC). In BFC, also known as freeze concentration by freezing–thawing, the liquid is completely frozen and the temperature of the centre of the product is set below the freezing point. Subsequently, the block is thawed and the concentrated fraction is then separated from the ice fraction. The separation process is occasionally assisted by external forces such as centrifugation or vacuum, since these external forces increase the efficiency of the procedure [5]. Previous studies have suggested that the suction during the vacuum process by a pump takes advantage of the channels between the ice crystals [6]. Therefore, the result is an increase in the extraction of the concentrate fraction from the ice fraction, and an improvement of the efficiency and solute recovery. The use of vacuum as an assisted technique to enhance the freeze concentration performance has been studied in saline solutions, sucrose, coffee extracts, red wine, orange juice, and blueberry juice [6,7,8,9,10,11,12]. In this case, BFC could be an alternative technology for the concentration of by-products, such as whey. In dairy processing, BFC offers to minimize the thermal damage on sensitive milk components, such as proteins and flavours, among others. Thus, it provides an opportunity for producing dairy ingredients with enhanced functional and organoleptic qualities.

In particular, FC has been applied on dairy products [2,3,13,14,15,16], and thus, for the concentration of whey, the maximum concentration ranged between 25 and 35 wt.% [2,16]. Moreover, there has been a growing number of studies on the concentration of dairy products through SFC technology (it comprises of a primary phase formation of ice nuclei (nucleation), followed by a secondary phase growth of ice nuclei in the solution) [14,17], by layer crystallization [18,19] and BFC assisted by gravity and microwave processes [2,3,15,16]. Until now, no study on the application of vacuum-assisted BFC applied to whey is known.

On the other hand, whey contains at least half of the total solids present in the initial whole milk, and hence, it can be considered as a valuable by-product with several applications, especially, in the food industry [20]. Recent research stated that the components of whey are difficult to degrade, creating a major problem to any wastewater treatment plants [21]. Therefore, the use of a concentration technique could be applied in the development of new products, and at the same time, it may help to solve the industrial waste problem [22]. BFC assisted by vacuum suction, due to its cheaper capital and operating costs than other FC alternatives could be an attractive technology for dairy industries [23].

Taking into account the above, the objective of this study in a first phase was to research the concentration of vacuum-assisted BFC of fresh cheese whey, by studying the influence of the initial concentration (C_0_), time (t), and vacuum pressure (V) on the response variables, concentration index (CI) and solute yield (Y). In a second phase, the best conditions of the previous stage (time and vacuum pressure) were applied to three types of whey, designated by: fresh cheese whey (CSW1), mató (a typical fresh cheese of Catalonia, Spain), cheese whey (CSW2), and blue cheese whey (CSW3). Additionally, the effect of vacuum assisted BFC on lactose content was also studied.

## 2. Materials and Methods

### 2.1. Material

Cheese whey was provided by a local supplier (Can Corder, Lliça d’Amunt, Barcelona, Spain) from cheese process. The three types of cheese was produced, namely, fresh cheese, mató, and blue cheese, use pasteurized cow’s milk as raw material.

CSW1 from fresh cheese. Its manufacture is very simple and consists of two stages: coagulation is essentially lactic (with ferment) and normally lasts 24 h, while draining is never excessive and is carried out in the container itself by dividing the coagulum into portions. Whey without salt.

CSW2 from mató: It is an enzymatic coagulation cheese (with rennet), drained by draining, lightly pressed (self-pressed) by periodic turning. Therefore, it is soft paste. No ripening. Whey without salt.

CSW3 from blue cheese: It is a matured cheese produced by enzymatic coagulation. Salty milk whey.

### 2.2. Freezing and Vacuum Suction Procedure

The experimental procedure of vacuum-assisted BFC was carried out according to Petzold et al. [8]. Firstly, whey samples (45 mL) were placed in plastic tubes (internal diameter: 22 mm) and were frozen at −20 ± 2 °C for 48 h in a static freezer (Fricon Model THC 520, Portugal). Due to our availability of freezing equipment, other different conditions have not been considered. The tubes were covered with thermal insulation made of polystyrene foam (8 mm thickness, thermal conductivity: K = 0.035 W/mK) in order to facilitate axial heat transfer. The frozen whey tubes were removed from the freezer and transferred to a suction stage. The suction was generated by connecting a vacuum pump (Comecta, Spain; Pump rate: 3.6 m^3^/h; Vacuum limit: 0.1 mbar) at the bottom of the frozen sample at room temperature (Figure 1). Subsequently, the concentrated solution was collected, and the concentration of the soluble solids within the solution as well as the concentration of the ice phase were measured by a refractometer (ATAGO DBX-55 Japan; Measurement range: 0.0–55 °Bx; Accuracy: 0.1 °Bx ± 0.1%). All measurements were taken in triplicate.

### 2.3. Experimental Design

#### 2.3.1. Effect of Factors on Response Variables

A full factorial design (FFD) was used to study the effect of the following three independent factors: initial soluble solids concentration (C_0,_ °Bx), time (t, min), and vacuum pressure (V, kPa), on the response variables: solid yield (Y) and concentration index (CI). The independent factors and their levels have been selected based on previous studies [6,11,12], and according to the initial tests carried out in our laboratory. The experimental design comprised eight combinations of the independent factors, as shown in Table 1. All tests were carried out in triplicate.

According to previous tests, 20 and 60 min under suction vacuum were adopted to ensure a sufficient concentrated sample, and to avoid vacuum break in the ice column. In this study the pressures of 10 and 70 kPa, are absolute pressures (absolute atmospheric pressure 101 kPa) that correspond to 90 and 30 kPa of vacuum pressure, respectively.

#### 2.3.2. Vacuum-Assisted BFC Tests on Three Types of Cheese Whey

In the second stage of experiments, the best conditions of time and vacuum pressure obtained in the previous phase with respect to the solute yield parameter (Y) were applied to the whey of three types of cheese: fresh cheese whey (CSW1), mató cheese whey (CSW2), and blue cheese whey (CSW3). Additionally, in these tests, the percentage of recovery of lactose with respect to the initial sample was determined.

### 2.4. Lyophilization Process

To carry out the freezing curves at different concentrations, as well as for the tests with whey at 19 °Bx, it was necessary to lyophilize the initial sample to obtain dry sample, which, once re-dissolved in water, allowed for the samples to be prepared at the desired concentration. Whey samples (approximately 25 mL) were frozen in Falcon tubes and placed horizontally in the freezer at −20 °C for 48 h. Then, the Falcon’s cap was removed, and plastic wrap was placed to facilitate sublimation. The frozen samples were placed in the lyophilizer (CRYODOS FD-10 Series, Telstar Industrial S.L, Spain; Vacuum pump nominal flow: 6 m^3^/h), for 48 h at −56.6 °C and a vacuum pressure of 4.7 × 10^−2^ mbar. Once the dried milk was obtained, the corresponding solutions were prepared with distilled water at 15, 25, and 35 °Bx. The same procedure was followed to prepare the samples at 19 °Bx.

### 2.5. Response Variables

Concentration index (CI) is defined as the relation between the concentration of soluble solids in the concentrated solution (C_f_), and the concentration of soluble solids in the initial whey (C_0_), according to Equation (1) [24]. The concentration index is also known as relative concentration [25].
(1)$$CI=\frac{C_{f}}{C_{0}}$$

Solute yield (Y) was calculated to analyze the soluble solids recovery. Y was defined as the relationship between the mass of soluble solids present in the separated liquid and the mass of soluble solids present initially in the initial solution. This can be seen in the following Equation (2) [26].
(2)$$Y\;\left(\%\right)=\frac{C_{f}\cdot m_{f}}{C_{0}\cdot m_{0}}\cdot 100$$
where C_f_ and C_0_ are the soluble solids content (°Bx) of the concentrate and initial whey, m_f_ and m_0_ are the concentrated and the initial whey mass (g).

### 2.6. Validation of Results

For each experiment, an ice mass ratio W_exp_ (kg ice/kg initial) can be defined, as the amount of ice obtained, with respect to the initial amount of sample in each experiment. In FC systems, ice handling can be complex and can be a source of error. Alternatively, this ratio can be estimated through a mass balance, depending on the concentrations of the ice, concentrated and initial whey sample. The measurement of concentrations is much simpler and more reliable than the data obtained from ice handling. It is denoted as W_p_, Equation (3) [27].
(3)$$W_{p}=\frac{C_{f}-C_{0}}{C_{f}-C_{i}}\cdot 100$$

The quality of the fit between experimental (W_exp_) and predicted (W_p_) values for N experimental points, i.e., the deviation between experimental and theoretical data, was tested by the Root Mean Square (RMS) as follows (Equation (4)).
(4)RMS%=100∑Wexp−WpWexp2N

### 2.7. Lactose Concentration

The lactose concentration of the initial, concentrated, and ice fractions were determined. The lactose content procedure was carried out according to Schuster-Wolff-Bühring et al. [28], with modifications. Hewlett Packard 1100 Series HPLC System (Agilent Technologies, Waldbronn, Germany) equipped with a Beckman 156 refractive index detector was used for determination. The separation was achieved with a tracer carbohydrate (250 × 4.6 mm, 5 μm) column (Teknokroma, Sant Cugat del Vallès, Barcelona, Spain). The volume injected was 20 µL and the mobile phase was a mixture of acetonitrile (Panreac Química SLU, Castellar del Vallès, Spain) and ultrapure water (75:25). The flow rate and column temperature were maintained as 1.3 mL/min and 28 °C, respectively. The detection was carried out with a refractive index detector, adjusting the zero against the mobile phase. Before the determinations, a portion of 1 mL samples was diluted with 8 mL of ultrapure water and mixed. Thus, 0.5 mL of Carrez Reagent 1 and 2 were added and mixed for 1 min. The mixture was allowed to settle for 15 min, and filtered by a nylon syringe filter (a pore diameter of 0.45 μm) (Agilent, Santa Clara, CA, United States). Each sample was prepared and injected in triplicate.

### 2.8. Statistical Analysis

The results obtained were statistically analysed using the application ‘Minitab 18’ for Windows (Minitab Inc., State Collage, PA, USA) and expressed as the mean ± standard deviation. To determine significant differences (*p* < 0.05) between results, one-way analysis of variance (ANOVA), and Tukey tests were used.

Full factorial design (FFD) with 3 factors and 2 levels (2^3^) was applied, with α = 0.05.

## 3. Results

### 3.1. Freezing Point Depression

The freezing point of a liquid depends on the concentration and type of solutes present in the solution [29], and thus, a high level of dissolved solids means a low freezing point. At the same mass concentration (%*w/w*), the solutes with low molecular weight (MW) have high molality, and therefore, a low freezing point. For cheese whey, the freezing point is influenced by the concentration of lactose, chlorides and other salts, and it is lower than that of pure water [30]. The freezing point was determined for initial and concentrates CSW1 whey with concentration of 15, 25 and 35 °Bx (Figure 2).

As expected, the freezing point determined through this experiment increased with a growing concentration of solids. The same trend was observed for whey [1,19,30] and milk [31]. For comparison, Figure 2 includes the results of salted cheese whey from the same manufacturer [19]. The trend line obtained is above the line of salted whey, and it may suggest a low concentration of salt in the tested whey. On the other hand, Figure 2 includes the freezing points of an ideal solution obtained from the Van’t Hoff equation, taking 235 g/mol as the effective molecular weight for whey milk, suggested by Baschi et al. [30]. As can be seen, there is a very good correlation between the experimental (CSW1) and the calculated value for ideal solutions. This can be very useful for future freeze concentration works.

### 3.2. Solute Yield and Concentration Index

The responses obtained for Y and CI from the eight experiments are shown in Table 2. The best results for Y were obtained at a vacuum pressure of 10 kPa, an initial concentration of 7.5 °Bx and vacuum time of 60 min. For the CI parameter, the best condition was also 10 kPa, 7.5 °Bx, and 20 min of vacuum. The values suggest that it would be possible to improve the Y parameter by increasing the vacuum time.

This is confirmed by the results of ANOVA presented in Table 3, since the *p*-values are shown in Table 3, indicating that all individual effects, as well as the interaction between C_0_ and t, were significant (*p* < 0.05) on the solute yield (Y) and the concentration index (CI).

### 3.3. Vacuum-Assisted BFC Tests on Three Types of Whey

Based on the analysis of the results obtained in Section 3.2, vacuum-assisted BFC process was performed in three different whey samples, under conditions that allow for reaching the best values of solute yield (Y). These conditions correspond to an absolute pressure of 10 kPa and a time of 60 min under suction vacuum. The tests were performed in triplicate resulting in a total of 9 tests. Under these conditions, the Y values were close to 70%, as shown in Table 4.

Additionally, Figure 3 shows HPLC lactose chromatograms for the initial and concentrated of CSW1 whey.

On the other hand, Figure 4 shows the lactose concentrations (g/L) in the initial whey, in the concentrate and ice fractions, of the three types of whey tested (CSW1, CSW2, and CSW3).

The results of the tests carried out with the three types of whey were validated, comparing the results calculated with the Equation (3) (W_p_), with the measures of ice mass ratio obtained experimentally (W_exp_). In Figure 5, the values W_p_ and W_exp_ of the 9 experiments are presented.

The RMS value for the experiments performed with the three types of serum (CSW1, CSW2, and CSW3) is 6.5%.

## 4. Discussion

The concentration index (CI) showed values greater than 1 in all cases, with a downward trend as time increased. This suggests that the concentrated extract is mainly collected in the first thawed fractions. Similar behavior has been reported by other studies [32,33] in defrosting sugar solutions. These results may indicate that the solute in the frozen sample is recovered not only with fusion, but also with a diffusion of solute from the freeze concentrate phase (enhanced with vacuum). For Y, the behavior is the opposite, since the longer the vacuum time, the greater the amount of solutes recovered, even if they have a lower concentration.

Specifically, the factor that most influences Y is the vacuum time (t), while for CI, the main factor corresponds to the initial concentration (C_0_). As can be seen in Table 2, the higher initial concentration acquired the higher concentrated solution, and in turn, the higher concentration solution obtained the higher increase in viscosity. According to Sánchez et al. [19], in the initial concentration range studied (7.5 to 19 °Bx), the viscosity of whey increased more than twice. In BFC, the capacity of ice separators is inversely proportional to the viscosity of the concentrate [32]. Therefore, as the viscosity increases, the efficiency and solute yield (Y) are reduced. Other authors [33] suggest that the increase of the initial concentration forms thinner and smaller ice structures. These small crystals of melted ice make it difficult to recover the concentrate fluid, and therefore, a low Y value is obtained with the highest concentration of whey. Table 3 shows that the C_0_•t interaction is significant for both Y and CI. This suggests that centrifugation duration (t) has a greater effect at low initial concentrations (C_0_) than at higher concentrations. This may also be related to the higher viscosity of the serum at higher concentrations, and therefore more difficult to extract under vacuum.

The freezing rate of the samples is also an important factor in the FC process. The freezing time of the 45 mL whey samples used in the BFC system has been estimated through applying the model proposed by Pham [34], obtaining an average value of 234 min, which is equivalent to an average freezing rate of 5.7 μm/s. This value is very close to previous works on BFC applied in orange juice [35]. In addition, this freezing rate is lower than the critical value (approximately 8 μm/s) provided by other authors [24,26]. These authors reported that for velocities higher than 8 μm/s, the ice occluded the solutes during the freezing stage, and it was not possible to expect a considerable separation of the concentrated solution from the ice fraction.

The vacuum-assisted BFC process has been applied on three types of whey (CSW1, CSW2, and CSW3) in a single stage, under the conditions that correspond to the best values of the previous Y tests (60 min and 10 kPa). The results obtained were 73%, 69%, and 71% for CSW1, CSW2, and CSW3, respectively. These results are better than those obtained in trials listed in Table 2 (65.5%). It may be due to the fact that in the previous trials the initial concentration (7.5 °Bx) was higher than that of the three dairy whey (between 5.7 and 6.6 °Bx), and as previously demonstrated, with a high initial concentration, there is a low yield of solutes. A work on PFC in whey [36] indicated a maximum of 76.4% (*w/w*) of solute yield in four freezing/thawing steps.

From Figure 4, CI values close to 3.5, 4.1, and 3.9 can be deduced for lactose of the three types of whey (CSW1, CSW2, and CSW3, respectively). A lactose concentration index of 3.5 has been reported by Korotkiy et al. [37] using a double FC system (two steps) in milk whey. The results of lactose CI in the present study are equal or higher than those of CI of soluble solids (Table 4), which seems to suggest that lactose tends to pass more easily to the concentrated phase than into the ice. Similar results have been informed by Aider and de Halleux [15], who studied the milk whey under BFC process in successive stages. In the first stage, the CI of the solids was 1.86, while the CI of lactose for that same stage was 2. This behavior may be due to the fact that lactose, a low molecular weight disaccharide compared to other whey molecules, mainly proteins, is easier to move and more difficult to trap in ice than other molecules with high molecular weight. This trend is consistent with Kawasaki et al. [38], since they found that lower molecular weight solutes were separated and concentrated more efficiently than higher molecular weight solutes. This corresponded well with the magnitude of the diffusion coefficient for each solute.

Results on BFC applied in whey have been reported, where thawing was performed by gravity [3,15], without the aid of vacuum. CI for solids and lactose around 2 were obtained. Therefore, the results of this study (CI between 3.5 to 3.8 for solids, and 3.5 to 4.1 for lactose) suggest that the application of vacuum in a single step can help to improve the recovery of the concentrated liquid fraction. A work of Lamkaddam et al. [36] related to the application of PFC in whey obtained a maximum content of 20.52% (*w/w*) of total solids in four steps. The CI of total solids increases in each stage of FC and varies from 1.57 in the first stage, up to 3.42 in the fourth stage.

Based on the CI values of the soluble solids and of the lactose obtained in the present study, it seems possible that using the vacuum-assisted BFC technique, at least 70% of the lactose contained in the initial sample can be recovered, with a high CI (3.5–4.0). This performance can be increased if the formed ice is thawed and subjected to a new vacuum-assisted BFC stage.

Despite the difficulty in handling ice, a good agreement was observed between the experimental (W_exp_) and predicted (W_p_) ice mass ratio, as shown in Figure 5. An RMS of 6.5% was obtained, lower than the limit of 25% suggested by Lewicki [39] to consider an acceptable fit. These values were close to those reported in previous studies [6,11].

## 5. Conclusions

The vacuum-assisted BFC technique has been applied to whey samples. The results show that the variables studied (initial concentration, pressure, and vacuum time) have a significant effect on the process. The best solute yield (Y) results were obtained at a pressure of 10 kPa, 7.5 °Bx of initial concentration and 60 min of vacuum, while for the concentration index response variable (CI), the best conditions were 10 kPa, 7.5 °Bx, and 20 min. In a second phase of the work, by applying the conditions that provide higher solute yield (10 kPa and 60 min) to three different types of dairy whey, Y values around 70% or higher are obtained in a single stage. In all three types of whey, the CI for lactose are somewhat higher than those for soluble solids, suggesting that lactose tends to remain in the concentrated phase rather than on ice. In this way it is possible to recover, in a single step, at least 70% of the lactose contained in the initial whey. Vacuum-assisted BFC technology is postulated in this way, as an alternative for the recovery of valuable components contained in whey. Finally, the results are positive for the parameters tested, but it is necessary more studies to establish the effectiveness of BFC in the treatment of cheese whey.

## Figures and Tables

**Figure 1 foods-12-00836-f001:**
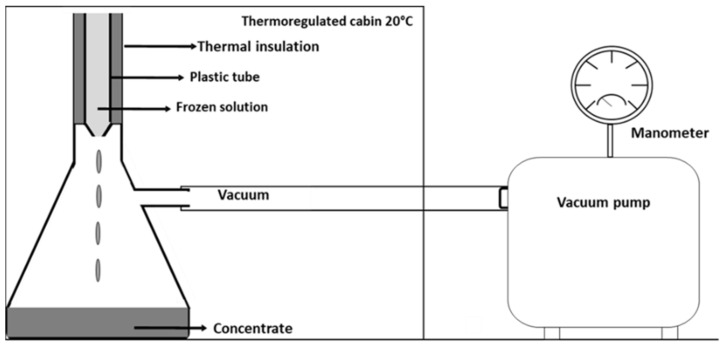
Vacuum unit for freeze concentration.

**Figure 2 foods-12-00836-f002:**
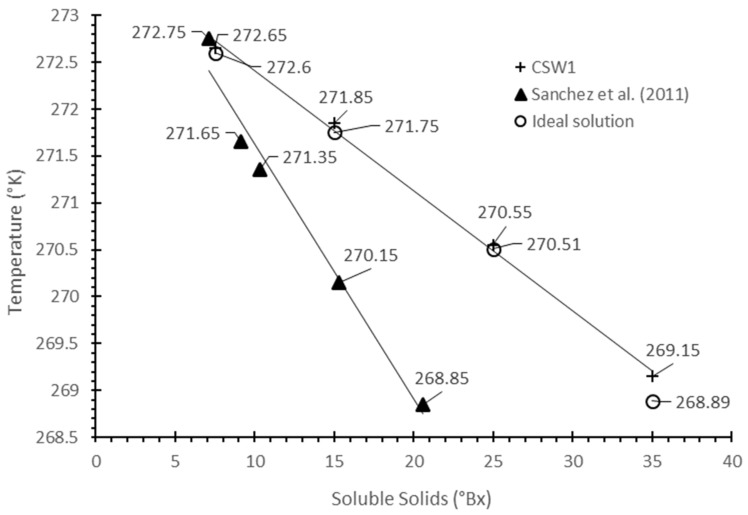
Freezing point (°K) of the CSW1 whey and an ideal solution, as a function of the soluble solids content (°Bx).

**Figure 3 foods-12-00836-f003:**
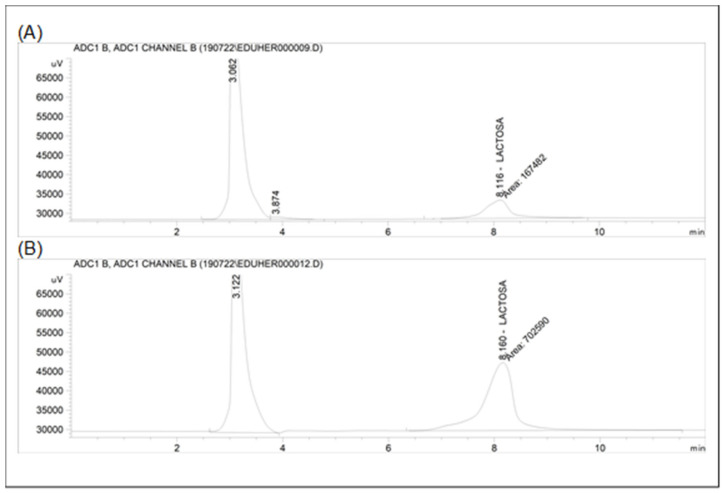
Lactose chromatograms of initial (**A**) and concentrated whey (**B**).

**Figure 4 foods-12-00836-f004:**
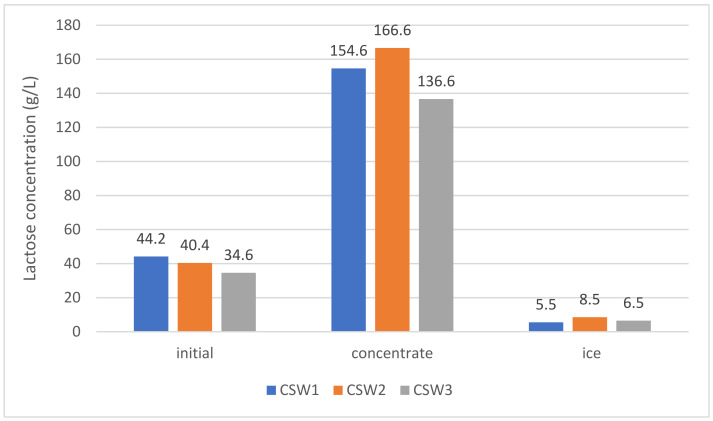
Lactose concentration of initial whey, concentrate and ice for three types of whey (CSW1, CSW2, and CSW3).

**Figure 5 foods-12-00836-f005:**
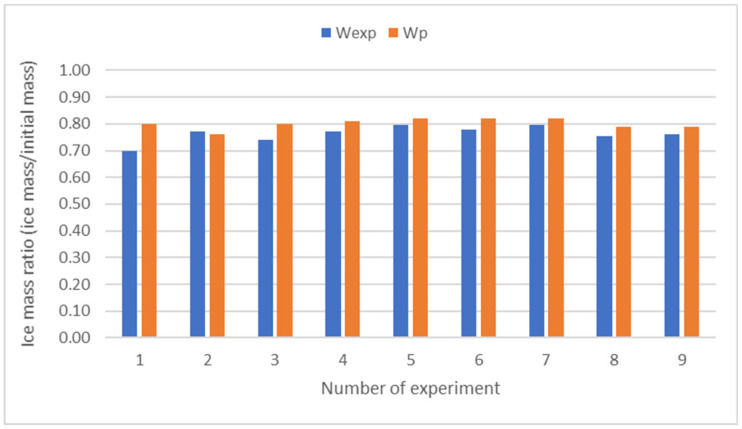
Experimental (W_exp_) and predicted (W_p_) values of ice mass ratio for CSW1 (experiment number 1, 2, and 3), CSW2 (experiment 4, 5, and 6), and CSW3 (experiment 7, 8, and 9).

**Table 1 foods-12-00836-t001:** Experimental design.

	C_0_ (°Bx)	t (min)	V (kPa)
Minimum	7.5	20	10
Maximum	19	60	70

**Table 2 foods-12-00836-t002:** Full Factorial Design (FFD), responses of solute yield (Y) and concentration index (CI). Within a column, different lowercase letters denote significant differences (*p* < 0.05) between Y and CI.

V (kPa)	C_0_ (°Bx)	t (min)	Y (%)	CI
10	7.5	20	28.4 ± 4.4 ^b^	5.3 ± 0.4 ^a^
10	7.5	60	65.5 ± 3.8 ^a^	3.4 ± 0.2 ^b^
70	7.5	20	9.4 ± 4.8 ^cde^	4.5 ± 0.2 ^a^
70	7.5	60	36.3 ± 10 ^b^	3.2 ± 0.7 ^b^
10	19	20	7.4 ± 3.9 ^de^	2.6 ± 0.1 ^bc^
10	19	60	24.3 ± 2.0 ^bc^	1.7 ± 0.1 ^cd^
70	19	20	0.26 ± 0.1 ^e^	1.2 ± 0.3 ^d^
70	19	60	21.9 ± 1.0 ^bcd^	1.5 ± 0.2 ^cd^

**Table 3 foods-12-00836-t003:** Results of ANOVA for solute yield (Y) and concentration index (CI). The asterisk (*) indicates a significant effect (*p* < 0.05).

	Y		CI	
	F-Value	*p*-Value	F-Value	*p*-Value
Vacuum (V)	39.08	<0.000 *	14.57	0.002 *
Concentration (C_0_)	86.03	<0.000 *	197.47	<0.000 *
Time (t)	123.44	<0.000 *	32.34	<0.000 *
V•C_0_	17.68	0.001 *	0.53	0.479
V•t	0.34	0.566	7.11	0.017 *
C_0_•t	7.57	0.014 *	16.79	0.001 *
V•C_0_•t	2.60	0.127	0.57	0.461

**Table 4 foods-12-00836-t004:** Results of concentration index (CI) and solute yield (Y) for the three types of serum subjected to 10 kPa for 60 min.

Whey Type	C_0_ (°Bx)	CI	Y (%)
CSW1	6.6	3.5 ± 0.1	73 ± 1
CSW2	5.7	3.8 ± 0.2	69 ± 1
CSW3	6.4	3.8 ± 0.2	71 ± 4

## Data Availability

Data is contained within the article.

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
