# Peer review of "Vacuum-Assisted Block Freeze Concentration Studies in Cheese Whey and Its Potential in Lactose Recovery"

_foods, 2023, doi:10.3390/foods12040836_

Round 1
Reviewer 1 Report
The manuscript describes an interesting experiment with use modern Vacuum Assisted Block Freeze Concentration technology, that uses of freezing water as a way to thicken liquid mixtures in the food industry. Used this method for the purpose of whey concentration, which is a heavy burden for the dairy industry, seems to be very promising. The manuscript is well written and the study is carefully conducted.
Still, some clarifications could improve the quality of the manuscript:
- Title: It is not adequate to the described main goal of the experiment. The recovery of lactose using the BFC technique was not the main and only goal of the research. This goal should be more emphasized throughout the paper or the title of the manuscript should be modified.
- Line 98-99: „Additionally, the effect of vacuum assisted BFC on lactose content was also studied.” Wasn't this the main subject of research as the title suggests?
- Line 103-104: should be „was produced”
- Line 229-239 (Figure 2): For which of the tested wheys (CSW1, CSW2, CSW3) was the freezing point determined? It is not indicated in the figure and description of the figure is not satisfactory?
- Line 264-269: “initial” or “original” whey? For readers’s comfort, uniform vocabulary should be used consistently throughout the text of the manuscript.
- Line 272-276: “Wp” or “Wpred” and “We” or “Wexp”? Please use uniform shortcuts to be consistent.
- Line 278: “RMS”, please explain the abbreviation. How the value RMS was calculated?
The important tip for the future works is that the references should be not older that 10 years. In this manuscript 40% references are older than 10 years.
Reviewer 2 Report
The article presents the benefits of the vacuum-assisted BFC technique in cheese whey samples. In general, the article "Vacuum-Assisted Block Freeze Concentration as a tool to recover lactose from cheese whey" is well written, with well-defined methodologies. The results show that BFC is an important alternative for recovering valuable components contained in whey, and has wide utility for the dairy industry and future research. However, a few considerations need improvement.
1. Line: 150-151: The phrase "a typical fresh 150 cheese from Catalonia, Spain" does not need to be here. It's been explained before.
2. Table 2 and Table 3: Statistical groups can be added to the results.
3. Conclusion: as this is unprecedented research and there are few references, it is important to conclude that the results are positive for the parameters tested, however, more studies are needed to evaluate other parameters.
Reviewer 3 Report
The manuscript looks at freeze concentration of whey streams. Out of the thousands of dairy science articles written very few would put much attention in to freeze concentration. Thus the topic has an element of novelty that I think would be well received by the readership. Personally I'm struggling with review, there isn't much science to address, the manuscript is really focusing on the innovation it self.
1) I did not understand how ice was being measured. was this determined by the relative increase in lactose concentration.
2) Freezing point depression is a colligative property that could be predicted reasonably simply. I'm not sure why comparisons isn't used in figure 2. Or perhaps I have not understood the aims?
3) Could the authors give an indication of approximate eutectic temperatures (I think this is what other readers might understand as glass transition temperature). I ask becuase I have a feeling that it might be so much below the operating temperature window it may be irrelevant. At least at low solids.
